# Characterization of Neutralizing Human Anti-Tetanus Monoclonal Antibodies Produced by Stable Cell Lines

**DOI:** 10.3390/pharmaceutics14101985

**Published:** 2022-09-20

**Authors:** Tania Maria Manieri, Daniela Yumi Takata, Roselaine Campos Targino, Wagner Quintilio, João Victor Batalha-Carvalho, Camila Maria Lucia da Silva, Ana Maria Moro

**Affiliations:** 1Biopharmaceuticals Laboratory, Butantan Institute, Sao Paulo 05503-900, Brazil; 2Interunits Graduate Program in Biotechnology, University of Sao Paulo, Sao Paulo 05508-270, Brazil; 3Graduate Program in Immunology, University of Sao Paulo, Sao Paulo 05508-270, Brazil; 4Center for Research and Development in Immunobiologicals (CeRDI), Butantan Institute, Sao Paulo 05503-900, Brazil; 5National Institute for Science and Technology (INCT/iii), University of Sao Paulo, Sao Paulo 05403-900, Brazil

**Keywords:** tetanus toxin, neutralization, GT1b, therapeutic antibody, vector construction, affinity, SPR, Fc gamma receptor, glycosylation

## Abstract

Tetanus toxin (TeNT) is produced by *C. tetani*, a spore-forming bacillus broadly spread in the environment. Although an inexpensive and safe vaccine is available, tetanus persists because of a lack of booster shots and variable responses to vaccines due to immunocompromised status or age-decreased immune surveillance. Tetanus is most prevalent in low- and medium-income countries, where it remains a health problem. Neutralizing monoclonal antibodies (mAbs) can prevent the severity of illness and death caused by *C. tetani* infection. We identified a panel of mAbs that bind to TeNT, some of which were investigated in a preclinical assay, showing that a trio of mAbs that bind to different sites of TeNT can neutralize the toxin and prevent symptoms and death in mice. We also identified two mAbs that can impair the binding of TeNT to the GT1b ganglioside receptor in neurons. In this work, to generate a series of cell lines, we constructed vectors containing sequences encoding heavy and light constant regions that can receive the paired variable regions resulting from PCRs of human B cells. In this way, we generated stable cell lines for five mAbs and compared and characterized the antibody produced in large quantities, enabling the characterization experiments. We present the results regarding the cell growth and viability in a fed-batch culture, titer measurement, and specific productivity estimation. The affinity of purified mAbs was analyzed by kinetics and under steady-state conditions, as three mAbs could not dissociate from TeNT within 36,000 s. The binding of mAbs to TeNT was confirmed by ELISA and inhibition of toxin binding to GT1b. The use of the mAbs mixture confirmed the individual mAb contribution to inhibition. We also analyzed the binding of mAbs to FcγR by surface plasmon resonance (SPR) and the glycan composition. Molecular docking analyses showed the binding site of an anti-tetanus mAb.

## 1. Introduction

Tetanus is an infectious, non-contagious, neuromuscular disease caused by the action of *Clostridium tetani* (*C. tetani*) exotoxin. *C. tetani* is a Gram-positive, spore-forming bacillus that is widespread in the environment [1,2]. Tetanus neurotoxin (TeNT) is one of the most potent neurotoxins, associated with a high lethality rate when no treatment is provided. It is synthesized as a single-chain protein of approximately 150 kDa and then cleaved by *C. tetani* proteases into a two-chain polypeptide, a heavy (H-heavy) chain of 100 kDa and a light (L-Light) chain of 50 kDa, linked by a disulfide bridge [3], which are essential for its toxicity [4]. The H chain is subdivided into two domains, the C-terminal (HC or fragment C) and N-terminal (HN), responsible for the functions of binding to the cell surface of neurons and translocation in the neural membrane, respectively. The L chain contains a zinc-binding motif and is responsible for the proteolytic action of the toxin [5,6,7]. TeNT affects the nervous system by blocking the release of the inhibitory neurotransmitters glycine and GABA (gamma-aminobutyric) in the synapses [6,8,9].

Despite the existence of a safe and low-cost vaccine, the lack of vaccination campaigns, booster doses, and variable response to vaccines due to an immunocompromised state or age mean that the incidence and mortality rate of tetanus are worrying [10,11]. In Brazil, the incidence rate decreased from 1.6 to 0.95 per million inhabitants in 2018, but lethality increased from 30.77 to 40.70%, mainly in the elderly and the northern regions [12]. Although tetanus is considered endemic in low- and middle-income countries, accounting for a significant mortality and disability rate in pregnant women, newborns, and the unimmunized or poorly immunized population [13], tetanus is everywhere, with 11,863 reported cases in 2020 (WHO). While in 2020, the African regions had a higher incidence (7.1), in 2021, the Eastern Mediterranean region had an incidence of 7.9 per 1,000,000 of the total population [14]. Vaccination is generally declining due to anti-vaccination groups, so health problems related to tetanus may increase. Spores of *C. tetani* are spread in the environment and can infect wounds caused by cuts, nails, needles, surgery, tattoo, piercing, etc.

As the damage caused by TeNT can be life-threatening, prophylactic treatments are necessary in cases of tetanus risk. As immunoglobulins cannot pass through the blood–brain barrier, they can only neutralize circulating toxin [15,16]. Based on clinical signs, a rapid passive immunization with equine anti-tetanus serum (ATS) or hyperimmune human tetanus immunoglobulin (TIG) is prescribed. Our group obtained a panel of human anti-tetanus monoclonal antibodies (mAbs) derived from memory B lymphocytes or plasmablasts isolated by single-cell sorting from the blood of vaccinated individuals [17]. Selected sequences were expressed by transient transfection in mammalian cells, and the mAbs, purified from the culture supernatant, were tested by ELISA (tetanus toxoid, tetanus toxin, fragment C), Western blot, and tetanus toxin binding to ganglioside GT1b inhibition assay. From the results, we chose five mAbs for the in vivo neutralization assay, individually or associated in pairs and a trio, the components of which showed binding to different domains of TeNT. Although the trio did not contain any of the two mAbs binding to fragment C, it showed efficient neutralization of TeNT by pharmacopeic animal testing (mouse neutralization test), demonstrating the importance of synergism for the complete neutralizing action. Some combinations of two mAbs could neutralize the toxin, although only at higher concentrations. The epitopes the mAbs bound to were identified by TeNT peptide microarrays [17].

To proceed with the development of human mAbs for TeNT neutralization that could be used later in clinical trials, we focused on obtaining permanent cell lines to produce each mAb in quantities and conditions appropriate for additional testing. Up to this development step, we had gathered information on the sequence of the variable region of the paired heavy and light chains and needed a platform to express complete IgGs. We ordered the synthesis of transport vectors containing the sequences of the constant regions of human heavy gamma 1 (IgG1) and light kappa chains (Κ), aiming to clone the variable sequences of each mAb into them. The variable sequences to be cloned came from human antibodies; so, they were naturally codon-optimized for mammalian cell expression.

This work aimed to advance in the stages of development of a mixture of human anti-tetanus monoclonal antibodies. This study involved the generation of cell lines, determination of culture conditions, analysis of the neutralizing capacity of the antibodies produced, and a preliminary evaluation of their pharmacokinetic behavior by analysis of the binding affinity to Fc gamma receptors (FcγR).

## 2. Materials and Methods

### 2.1. Vectors and Variable Sequences

Transportation vectors containing sequences coding for the constant regions [18] of the human gamma 1 heavy and kappa light chains were synthesized by GeneArt (Life Technologies, Carlsbad, CA, USA). The synthesis of sequences encoding the variable regions of the heavy and light chains of two mAbs, TT-117 and TT-120, was also ordered with the optimization of codons for expression into CHO-S cells (Gibco, Carlsbad, CA, USA). The variable sequences of the mAbs were deposited at GeneBank and received accession numbers from OP373409 to OP373418 related to the heavy chain and light chain of mAbs TT-117, TT-120, TT-140, TT-143, and TT-243.

### 2.2. Cloning

All the restriction enzymes were purchased from NEB (New England Biolabs, Ipswich, MA, USA). In the below description, heavy and light chains are referred to as H and L, respectively, and the letters C and V represent constant and variable, respectively. phuHC, TT-117_HV, and TT-120_HV were digested with AgeI-HF and SalI-HF; phuLC, TT-117_LV, and TT-120_LV were digested with AgeI-HF and BsiWI. The variable regions were cloned into the respective phuHC and phuLC, giving rise to phuHC_HV and phuLC_LV, containing variable and constant regions. Freedom^TM^ pCHO 1.0 expression vector (Gibco) and phuLC_LV were digested with EcoRV and PacI. The complete light chain was cloned into pCHO 1.0, giving rise to pCHO_LC_LV. pCHO_LC_LV and phuHC_HV were digested with AvrII and BstZ17I. The released heavy chain fragment was cloned into pCHO_LC_LV. All digestions were carried out for 4 h at 37 °C except for BsiWI, the best temperature of which was 55 °C. All digestion steps were assessed by 0.8% agarose gel electrophoresis. Before cloning, the digested vectors and inserts were purified using the Wizard^®^ SV Gel and PCR Clean-UP system kit (Promega, Madison, WI, USA). The ligation steps were performed with 25 ng inserts, 75 ng purified linearized vectors, and 1U of T4 DNA ligase (Invitrogen, Carlsbad, CA, USA). The reactions were incubated at 23 °C for 1 h, except for the ligation of heavy chain into the pCHO_LC_LV vector, which was incubated at 14 °C for 16 h. One Shot™ TOP10 Chemically Competent bacteria (Invitrogen) were transformed by heat shock. For this, 4 µL of the ligation product and 20 µL of the bacterial suspension were used. The suspension was plated on an LB agar plate with kanamycin and kept at 37 °C for 16–18 h. The presence of the insert was checked by colony PCR. Amplification was confirmed by 2% agarose gel electrophoresis stained with SyBR Safe (Invitrogen Carlsbad, CA, USA). Colonies with inserts were inoculated in 4 mL of TB medium with 50 µg/mL kanamycin (Gibco, Carlsbad, CA, USA) and incubated at 37 °C, 200 rpm, for 16–18 h. According to the manufacturer’s instructions, the supernatants were used to obtain the purified vector using the Wizard^®^ Plus SV Miniprep DNA Purification kit (Promega, Madison, WI, USA). The obtained vectors were checked by Sanger sequencing.

### 2.3. Expression and Purification

The complete pCHO vectors inserted with each mAb light and heavy chain were linearized by digestions with 15 µL of NruI enzyme for 50 µg of DNA. The digestion proceeded for 6 h at 37 °C. CHO-S cells (Gibco, Carlsbad, CA, USA) were transfected with the vectors for antibody expression. The procedure for stable transfection and the selection steps were performed with a Freedom ^TM^ CHO-S^TM^ Kit (Gibco, Carlsbad, CA, USA), following the manufacturer’s guidelines. At the end of each selection stage, aliquots of cells were frozen in liquid nitrogen. In four conditions, puromycin (Gibco, Carlsbad, CA, USA) and methotrexate (MTX) were used as selection agents. The pools were cultivated in a fed-batch for 14 days in Dynamis medium (Gibco, Carlsbad, CA, USA) to assess the specific productivity. The productivity of each stable pool was verified by glucose supplementation on days 3, 5, and 7. The mAbs were purified by affinity chromatography using protein-A Sepharose [17]. Purified protein was quantified by absorbance at 280 nm.

### 2.4. Fed-Batch Cultivation

Each mAb was produced by cultivation of the cells in 30 mL of Dynamis medium in shaker flasks incubated at 37 ºC, 8% CO_2_, 150 rpm until day 14 or less if viability dropped to 50%. Samples were collected on days 0, 3, 5, 7, 10, 12, and 14 to determine the cell density and viability (Vicell, Beckman Coulter, Indianapolis, IN, USA) and titer (SPR). Each mAb was purified by standard protein A chromatography (Cytiva, Upsalla, Sweden).

### 2.5. ELISA

ELISA microplates were coated with 100 μL of TeNT 5 μg/mL in PBS at 4 °C overnight. After blocking with 1% BSA (RT, 2 h), candidate antibodies were added (2-fold dilutions, from 50 to 0.4 ng/mL, in PBS with 1% BSA) and incubated (37 °C, 1 h). The plate was washed with 0.05% Tween 20 in PBS (PBS-T) before incubation with peroxidase-conjugated goat anti-human IgG antibody (Southern Biotech, Birmingham, AL, USA). TMB 3,3′,5,5′-tetramethylbenzidine (Sigma-Aldrich, Darmstad, Germany) at 100 µg/mL and 0.0045% hydrogen peroxide (Millipore, Burlington, MA, USA) in acetate/citrate buffer, pH 6.0 was used as substrate. The reaction was stopped with 2 M H_2_SO_4,_ and the absorbance was recorded at 450 nm. Serum from a vaccinated donor (approval by CEP/ICB/USP by Plataforma Brasil, number 1.515.313) was used as a control.

### 2.6. Tetanus Toxin Binding to Ganglioside GT1b Inhibition Assay

This test was performed as previously described [17]. Briefly, each mAb was mixed with TeNT and transferred to a plate coated with GT1b. An equine anti-tetanus serum conjugated with peroxidase detected the toxin bound to the plate, which was revealed with TMB.

### 2.7. Determination of mAbs Titer by SPR

To quantify mAb production during fed-batch culture, a BIAcore T200^®^ (Cytiva Healthcare) equipped with a CM5 sensor chip with protein A was used. Samples were collected on days 0, 3, 5, 7, 10, 12, and 14 of culture. A standard curve of a purified human monoclonal antibody was used to determine the mAb concentration following the standard BIAcore control software quantification routine. In brief, samples were applied for 180 s (10 µL/min), and after dissociation of 60 s (30 µL/min), the sensor surface was regenerated with 10 mM Gly solution (pH 1.7, 30 s, 30 µL/min). BiaEvaluation Software V 3.0 (Cytiva, Marlborough, MA, USA) was used to analyze the results.

### 2.8. Steady-State and Kinetic Affinities

All affinity studies were run on a BIAcore T200^®^. They were carried out at 25 °C in HBS-EP running buffer (0.01 M Hepes pH 7.4, 0.15 M NaCl with 3 mM EDTA and 0.005% Tween 20). BiaEvaluation Software V 3.0 was used to analyze the sensorgrams. The mAbs were immobilized on the CM5 sensor chip by amine coupling following the manufacturer’s instructions at around 700 RU. Then, 4-fold TeNT dilutions ranging from 100 to 0.39 µg/mL were injected for 600 s at 5 µL/min to determine the steady-state affinity. After 60 s of dissociation, the sensor chip surface was regenerated with a 10 mM Gly solution (pH 1.7, 30 s, 30 µL/min). Kinetic studies were carried out in a single cycle using 5 TeNT concentrations from 100 to 1.23 μg/mL (dilution factor 3) for 90 s at 30 μL/min. After 36,000 s of dissociation, the sensor chip surface was regenerated by a 15 μL pulse of 10 mM Gly pH 1.7. The kinetic constants were calculated using a Langmuir 1:1 interaction model with the average of two independent experiments.

### 2.9. Evaluation of Binding of Anti-Tetanus mAbs to Fc Receptors

mAbs binding to Fcγ and FcRn (neonatal Fc receptor) receptors was evaluated by SPR on a BIAcore T200^®^. The assays were carried out at 25 °C. BiaEvaluation Software V 3.0 was used to analyze the sensorgrams.

FcγRI: the purified antibodies’ affinities to FcγRI were evaluated by single-cycle kinetics. NTA Sensor (Cytiva Healthcare) was equilibrated in HBS-N buffer (0.01 M Hepes pH 7.4 and 0.15 M NaCl) and sensitized with 500 mM NiCl_2_ for 60 s at 10 μL/min. FcγRI (SinoBiological) at 0.5 μg/mL was captured for 120 s (10 μL/min). Five different sample concentrations ranged from 0.247 to 20 μg/mL (3-fold dilution), 240 s of contact, and 900 s of dissociation (30 μL/min). The sensor chip surface was regenerated with a 30 µL 0.25 M pH 8.5 EDTA pulse.

FcγRIIa/b and FcγRIIIa/b: Receptors (FcγR extracellular domain proteins, SinoBiological) were immobilized on a CM5 sensor chip (Cytiva) by amine coupling as per the manufacturer’s instructions. Five different sample concentrations in running buffer (HBS-EP-0.01 M Hepes pH 7.4, 0.15 M NaCl, 3 mM EDTA, 0.005% Tween 20) were used, ranging from 0.247 to 20 μg/mL (3-fold dilution), 270 s of contact, and 900 s of dissociation (30 μL/min). The sensor chip surface was regenerated between cycles, with a pulse of 15 µL 10 mM NaOH.

FcRn: Neonatal receptor, (SinoBiological) was covalently bound to a CM5 sensor chip surface by amine coupling. Serial dilutions of mAbs from 150 to 4.69 µg/mL (2-fold dilution) in running buffer (HBS-EP-0.01 M Hepes pH 6.0, 0.15 M NaCl, 3 mM EDTA, 0.005% Tween 20) were injected sequentially (association and dissociation 120 s, 30 µL/min) in a multicycle approach to assess the steady-state affinity. The sensor chip surface was regenerated between cycles with two pulses of HBS-EP pH 7.4 (17.5 µL/pulse).

### 2.10. mAbs Glycan Profile

A LabChip^®^ GX II instrument was used with Glycan Profiling Assay Reagent and Glycan Release and Labeling kits and the associated High-Resolution Protein LabChip (PerkinElmer, Waltham, MA, USA) to assess the mAbs glycan profile. The assays were performed according to the manufacturer’s instructions. Briefly, 10 to 60 µg of protein was denatured, and N-linked glycans were released enzymatically using PNGase F. After that, the released glycans were labeled with a fluorescent dye, separated, and detected by microchip-based capillary electrophoresis. The data were evaluated with the LabChip^®^ GXII software using standard glycans as controls. The result is expressed as a percentage of each glycan among those that the equipment can identify, namely, G0, G0F, G1, G1F, G2, G2F, SA, A2, Af2, and Mn5.

### 2.11. Molecular Docking

The in silico methodology was based on homology modeling of the antibody’s three-dimensional structure using the Pigs Pros server [19], followed by two rounds of protein-protein docking. The first round was performed in the ZDOCK server [20] to obtain the antigen–antibody complex, and the second round was performed with SnugDock [21]. The initial structure of tetanus toxin was a closed conformation (PDB code 5N0B). The extended conformation was obtained from the initial structure processed with the Morph Conformations tool of CHIMERA software [22]. Modeller was used to model the refinement of nonexistent loops in CHIMERA (v2.2.4) [23], and 3D representations were built using PyMol 2.5 [24].

## 3. Results

### 3.1. Generation of Cell Lines

The cloning of human anti-tetanus mAbs into the expression vector pCHO 1.0 was accomplished by sequential steps, as depicted in Figure 1. First, the variable light chain gene was ligated to the kappa gene in the transportation vector phuLC. The complete light chain was cloned into one pCHO 1.0 expression cassette (pCHO_LC_LV). The heavy chain followed the same route, later cloned into the second cassette of pCHO_LC_LV for the expression of complete IgG antibodies. After cloning, the heavy and light chain inserts were confirmed by horizontal electrophoresis. Sanger sequencing showed no mutations. Vectors comprising the five sequences were individually transfected into CHO-S cells and submitted to an increased concentration of puromycin and methotrexate, generating four stable pools each, identified as 1A, 1B, 2A, and 2B (28 pools in total: 20 for the 5 original mAbs and 8 for 2 of the mAbs that were codon-optimized). All the pools were tested in batch cultures, comparing the cell density, viability, and antibody yield. In the end, antibodies produced by pool 2B (more stringent selection) were chosen for subsequent analyses.

Each mAb was produced by cultivation of the pools in shaker flasks with Dynamis medium for 14 days in fed-batch conditions. Samples were collected on days 0, 3, 5, 7, 10, 12, and 14 to assess cell density, viability, and titer and used to calculate IVCC (integral of viable cell concentration) and Qp (specific productivity). Figure 2 shows the cultivation results in comparison with the five mAbs (original sequences only), indicating viable cell density and, viability (Figure 2A), IVCC (Figure 2B), titer (Figure 2C), and Qp (Figure 2D). It was possible to verify a difference in the expression of antibodies in the pools of different mAbs, considering that the cell density (Figure 2A) and IVCC (Figure 2B) were similar. Qp calculation reinforced the differences in mAb expression by different antibody sequences, with TT-243 and TT-140 being higher and lower producers, respectively (Figure 2C and Figure 2D). After the 14-day cultivation period, the supernatant of each of the five mAbs was collected, and the antibodies were purified by affinity chromatography with Protein A resin. SDS-PAGE was run under non-reducing and reducing conditions, showing integrity for each mAb.

### 3.2. Binding and Kinetics Characterization

The binding of purified mAbs to TeNT was tested by ELISA (Figure 3), showing no significant differences between them at the concentrations tested (Kolmogorov–Smirnov non-parametric test). For KD(M) calculation, immobilization of TeNT in CM5 sensors for the capture of mAbs was attempted first. The results were inconsistent, probably due to the toxin size and diversity in the availability of epitopes to which the five mAbs bind. Instead, each mAb was immobilized in sensor chips through a protein A capture. In this case, TeNT was the kinetic affinity assay’s ligand. No dissociation was observed with the standard incubation time (600 s). Extending the time to 36,000 s, the maximum allowed by the equipment, some mAbs, TT-243, TT-143, and TT-117, could still not dissociate, and KD(M) could not be calculated (Table 1). As it was impossible to observe the dissociation of some of the mAbs, as part of their characterization, we performed a steady-state affinity study with each mAb immobilized in a protein A sensor (Table 1). Figure 4 shows the kinetics graphs obtained by SPR with increasing concentrations of TeNT in a single cycle.

### 3.3. GT1b Toxin Binding Inhibition Test

The mAbs, individually (Figure 5A) and in mixtures (Figure 5B), were evaluated for their ability to inhibit the binding of TeNT to the GT1b ganglioside. Confirming previous results, even at higher concentrations, only mAbs that bind to fragment C (Figure 5A) could inhibit the binding to GT1b when used alone. Some mixtures showed binding inhibition of GT1b at different levels. However, this response seems to be associated with TT-117 and/or TT-140 in the mixture (Figure 5B). The estimated IC_50_ was 2.4 × 10^−8^ M for both individual TT-117 and TT-140 (5A) and increased to 7.2 × 10^−8^ M when combined (5B). The mixtures containing TT-117 presented IC_50_ between 10^−7^ and 10^−6^ M while the mixtures containing TT-140 presented IC_50_ at 1 × 10^−6^ M. The molecular docking of mAb TT-140 (Figure 6A,B) identified that mAb TT-140 interacts with residues close to the R pocket, such as TYR 1229 and ASN 1230. The details of this interaction are shown in Figure 7.

### 3.4. Evaluation of the Binding of Anti-Tetanus mAbs to Fc Receptors

SPR was used to evaluate the Fcγ receptor’s binding and affinity to characterize the produced mAbs:

FcγRI—There is no difference in the affinities obtained for the mAbs tested.

FcγRIIa—In this case, there is a slight difference in the affinity between the mAbs. It is interesting to note that only the clone TT-117 showed some interaction with the receptor on the sensor, with the affinity calculated in the order of µM. As the interaction of the other mAbs with the receptor on the sensor was lower, it was impossible to reliably estimate the values of the affinity constants under these experimental conditions.

FcγRIIb—Behavior similar to that observed with the FcγRIIa receptor. However, in this case, none of the mAbs showed interaction with the receptor in the sensor to allow the calculation of the kinetic constants.

FcγRIIIa and FcγRIIIb—Similar behavior was observed. None of the mAbs showed interaction with the receptor on the sensor to allow the calculation of kinetic constants.

FcRn—Despite the low interaction with the receptor immobilized on the sensor, no difference was observed between the steady-state affinities calculated for the different clones, indicating potentially similar pharmacokinetics behavior.

### 3.5. mAbs Glycan Profile

The glycan profile was assessed by capillary electrophoresis in a chip by using Caliper Labchip^®^ (Table 2). Among the glycans that the method identified, those fucosylated represented 57.66 to almost 75% of the total identified glycans. On the other hand, those of the high-mannose type represented approximately 3 to 9%, and those galactosylated varied from 4.57% up to 11.45%.

## 4. Discussion

The search for therapeutic anti-tetanus antibodies to replace human immunoglobulin and equine serum has been the subject of recent publications, demonstrating an increased interest in this therapeutic approach [25,26,27,28]. In common among them is the use of B cells from immunized individuals.

The methodology for obtaining human monoclonal anti-tetanus antibodies that was proposed previously [17] resulted in a panel of clonally related antibodies. These antibodies were screened and characterized in vitro to predict which ones should be tested in vivo and determine the best candidates for future therapeutic use. To neutralize TeNT, multiple epitopes must be blocked [29]. The mixture of mAbs has a combined or synergistic effect that could be explained by the fact that blocking multiple epitopes increases the chance of interference on various residues critical for toxicity and could favor the formation of larger immune complexes eliminated by phagocytosis. Two groups independently discovered new anti-tetanus mAbs capable of neutralizing TeNT when used alone [27,28] while other reports support the composition of mAbs to achieve full protection. The mixtures of three mAbs improved the survival rates among animals, and a mixture of 35 mAbs completely neutralized TeNT, showing that a combination of mAbs directed to different TeNT regions increased protective activity [26]. Anti-TeNT nanobodies also worked better as multimers [25].

The neutralizing composition of the three mAbs that we propose is interesting because each one binds to a different domain of TeNT without competition among them [17], showing that other mechanisms in addition to blocking the binding of TeNT to GT1b are necessary for neutralization. An mAb binding an epitope of HC-C was recently reported to neutralize TeNT by inhibiting the conformational change of TeNT at a low pH, blocking the membrane translocation step of the TeNT mode of action [27]. Another group obtained three individually neutralizing mAbs binding to distinct domains of TeNT, with partial competition between two of them [28]. The best neutralizer of this group, TT0069, needs 2.22 µg of mAb to prevent death. In comparison, our composition only needs 0.63 µg of total mAbs (1:1:1). In our work, the mixture of three mAbs guaranteed the survival of all ten mice in three consecutive dilution doses, and combinations of two mAbs increased protection at higher doses [17]. A direct comparison between the neutralizing capacity of the reported anti-tetanus mAbs is difficult due to differences in the in vivo assays, notably the number of mice in each study and the reagents used. We followed the neutralization test described by Pharmacopeia animal testing, which requires a fixed volume of TeNT, using the Limes paralyticum/10 (Lp/10) dose. Published neutralizing human anti-tetanus mAbs used TeNT measured by the lethal dose (LD_50_) [28] or ng/kg [27]. Although the assays did demonstrate consistency between them, a direct comparison would require the mAbs to be tested with the same reagents and conditions.

As for affinity kinetics, three of our mAbs (TT-117, TT-143, and TT-243) showed irreversible binding as they did not dissociate even at 36,000 s. TT-120 and TT-140 presented KD at 10^−10^ M, comparable to other published mAbs (10^−8^ M for TT0067, 10^−10^ M for TT069 [28], and 10^−9^ M for TT-110 [27]). Only one (TT-104) presented KD in the range of 10^−12^ M [27]. We measured the kinetics by capturing the mAbs on the CM5 chip, as did others. We decided to run the kinetic affinity assay on a single cycle, increasing the ligand concentration, which is recommended with longer dissociation times or when the ligand is sensitive to the regeneration conditions. Considering that TeNT can bind to receptors and be internalized by neurons and that, after internalization, TeNT is not reachable, anti-tetanus antibodies must bind to circulating TeNT and remain bound to them to neutralize the action of the toxin. A low dissociation rate implies a high avidity, leading to a stable interaction and irreversible binding, pharmacokinetics favorable for clinical use.

The entry of TeNT into neurons is preceded by the binding of fragment C of the toxin to gangliosides present in the membrane of these cells. As expected, those that bind fragment C of TeNT were able to inhibit 100% of the binding of TeNT to ganglioside GT1b. The high-affinity binding of the toxin to the gangliosides depends on the occupation of two sites (R and W) of the ganglioside [30]. Although mAbs TT-117 and TT-140 do not bind to these sites, inhibition may occur due to the conformational change in the toxin or to steric impairment caused by binding to the mAbs [30]. The non-occupation of these sites may explain why there was no in vivo protection when used as single antibodies [17]. However, when the in vivo testing was carried out, stable cell lines were unavailable, and TeNT-GT1b inhibition did not reach 100%, as we showed in the current work. Considering the inhibition assays, both TT-117 and TT-140 mAbs showed similar IC_50_, calculated as 2.41 × 10^−8^ M and 2.48 × 10^−8^ M, respectively. Interestingly, when the two mAbs were combined, the IC_50_ increased to 7.2 × 10^−8^ M, suggesting that a steric hindrance may impact the outcome. Of note is that when TT-117 is a component of different mAb mixtures, less mAb is needed to achieve IC_50_ than when TT-140 is present, in the range of 10^−6^ M. Concerning the two mAbs (TT-117 and TT-140) that inhibit the binding of TeNT to GT1b, competition between them was not fully elucidated, and other assays are needed. The binding of one seems to impair the binding of the other. We assumed this to be the steric hindrance hypothesis, as they bind to different epitopes (through the peptide array). We do not know whether one of these mAbs can dislodge the other, only that TT-117 is a stronger binder due to its kinetic affinity behavior. The linear peptide array indicated the epitope 1119–1128 is a ligand for TT-117, away from the R and W pockets. The in silico modeling showed that TT-140 partially overlaps the TT-117 binding region predicted by the array. LigPlot+ analysis showed a hydrogen bond between E1127 and S100A (CDR3-HC), which may inhibit the binding of TT-117 to that site. Further assays, in vitro and in vivo, are required to explore the competitive interaction between TT-117 and TT-140. They belong to different VDJ families and have completely different CDRs; TT-140 has a long CDR3H with 18 amino acids.

Only TT-140, from the five mAbs described here, presented a strong binding to an epitope using the peptide array technique [17]; the other four showed weak interactions, which could mean that they bind to conformational epitopes, as reported for other anti-tetanus mAbs. Perhaps this is the reason why a single mAb neutralized the toxin by binding to different regions of TeNT. Although the neutralization capacity of the trio of mAbs did not impact the binding of TeNT to GT1b, they may impair the binding of TeNT to a protein receptor on neuronal cells [31], an aspect that will be explored in further assays.

From the panel that we obtained, we selected five mAbs tested in vivo: two mAbs binding to the recombinant fragment C and three other mAbs binding to different TeNT regions (light chain, HC, and HN). To advance in the development stages of mAbs suitable for additional testing and future clinical trials, the present work focused on generating stable cell lines while constructing vectors that provide autonomy to generate permanent cell lines for a higher number of sequences. We used the sequences identified directly from B cells. As the host cell—CHO—is of hamster origin, we also ordered the synthesis of codon-optimized sequences for two mAbs: TT-117 and TT-120. When comparing mAbs produced by sequences assembled from the original amplification with those from the optimized synthesis, we were able to notice a slight difference in the production rate, which was higher when the sequences were optimized. This difference in titer does not seem to be related to the integral concentration of viable cells but the specific productivity. However, the differences implied no significance, evaluated by Kolmogorov–Smirnov statistical testing, and were not shown. This lack of significance is not unexpected, since mAb sequences of human origin are prone to expression in mammalian cells. The functional assays did not show differences between the original and codon-optimized sequences. Some mAbs express better, and TT-243 surpasses the others in production, likely due to its sequence. Cellular growth along the time was similar to all mAbs, as verified by the IVCC calculations.

In addition to binding to the target antigen, Fc-mediated functions are essential features of therapeutic mAbs that can affect the efficacy and safety. Fcγ receptors (FcγRs) play a critical role in Fc-mediated mAb functions, and the best known are antibody-dependent cellular cytotoxicity (ADCC), antibody-dependent cellular phagocytosis (ADCP), and complement-dependent cytotoxicity (CDC) [32]. Interactions with FcRn will interfere with mAb pharmacokinetics, improving or reducing their clearance and, therefore, their half-life in vivo [33]. In this study, within the experimental conditions used, there was no significant difference in the affinity constants between the mAbs and the FcRn receptor, indicating a similar pharmacokinetic behavior. On the other hand, the interaction with the other receptors was slightly different. The mAbs tested showed high affinity, in the order of nM for FcγRI. However, the interaction was weak with the other receptors, except for TT-117 for FcγRIIa and TT-120 and TT-140 for FcγRIIIa.

The glycan composition is a determining factor in the interaction of antibodies with receptors and their effector functions [34]. Under our experimental conditions, it was possible to observe a higher percentage of fucosylated glycans, as expected, and a lower amount of antibodies with high mannose moieties [35,36]. On the other hand, although the method adopted has limitations, it was possible to identify differences between the mAbs, which may eventually be due to cultivation conditions [36]. Since the glycan composition strongly influences the binding of the antibody to the Fc receptor, further cell culture optimization and more refined analysis of the glycan composition of these antibodies need to be performed after the cloning of the pools.

Compared to other mAbs or nanobodies described recently, there is no similarity in terms of the CDR3 sequence between our mAbs and others, confirming the singularity of the mAbs found in our panel. This also shows that a diversity of antibodies may result in protection against the tetanus toxin. We plan to explore the anti-tetanus mAbs described herein in the direction of a therapeutical proposal.

## Figures and Tables

**Figure 1 pharmaceutics-14-01985-f001:**
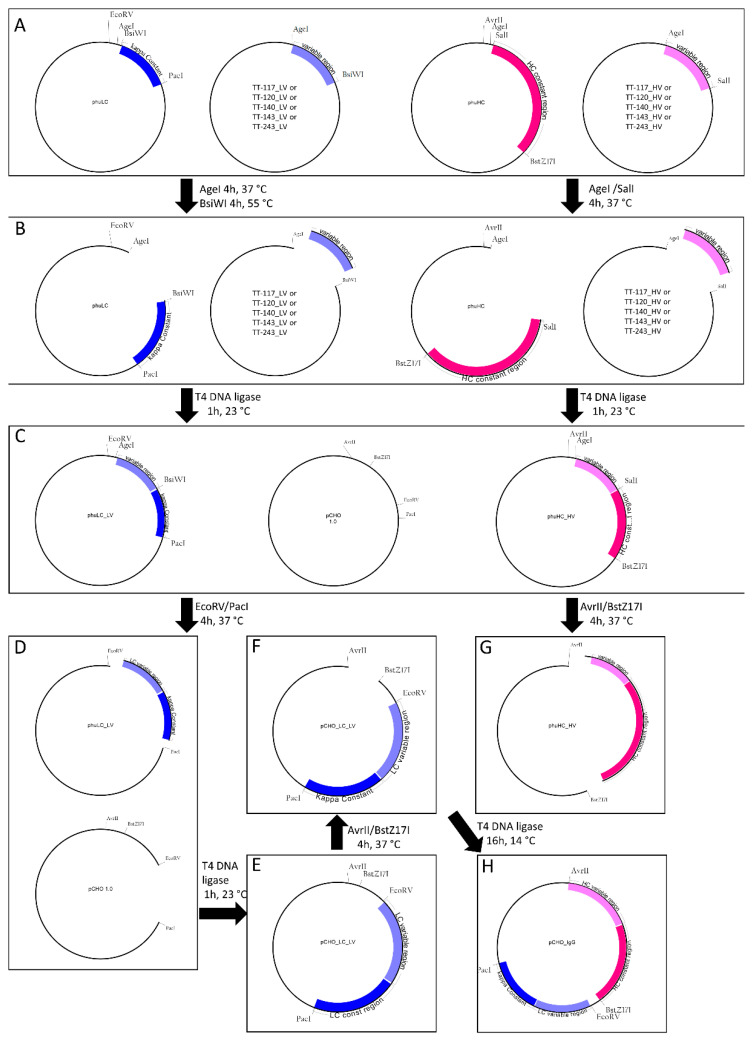
Construction of the expression vector. (**A**) Transportation vectors synthesized by GeneArt contain constant regions and vectors containing variable regions. (**B**) Digestion by AgeI-HF and SalI-HF for heavy chain cloning and Age-HF and BsiWI for light chain cloning. (**C**) Ligation results in vectors with a complete heavy or light chain. (**D**) Digestion with EcoRV and PacI for light chain release and preparation of vector pCHO1.0 for cloning. (**E**) Ligation of complete light chain released in D into pCHO 1.0. (**F**) Digestion with AvrII and BstZ17I of pCHO_LC_LV. (**G**) Digestion with AvrII and BstZ17I for complete heavy chain release. (**H**) Ligation of released complete heavy chain into linear pCHO_LC_LV and the complete expression vector.

**Figure 2 pharmaceutics-14-01985-f002:**
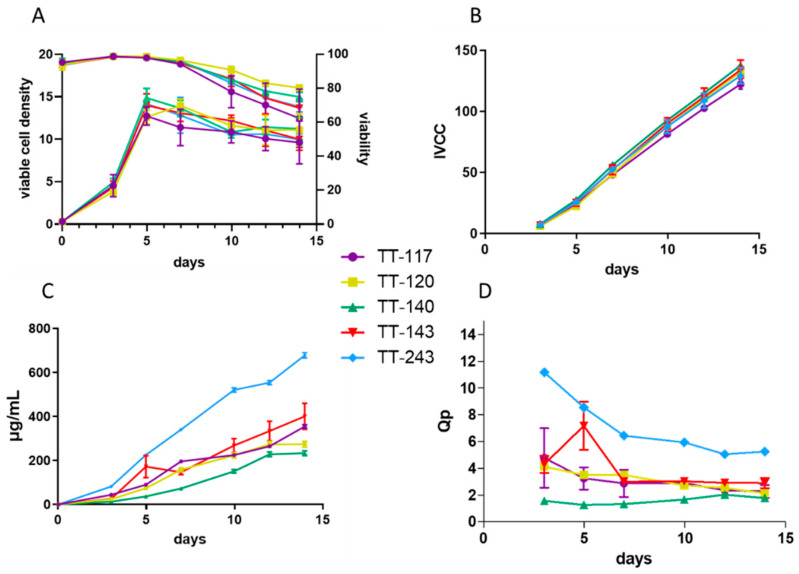
Graphs of fed-batch cultivation of stable transfection. Pools of mAbs were cultivated in a fed-batch mode for 14 days, with the evaluation of cell density and viability (**A**), IVCC (**B**), titer (**C**), and Qp (**D**). Each of the five mAbs is identified by the same color line. Graphs A and B, related to cell growth, show the mean and SD of data obtained by the four conditions of the selection agents for each mAb. Graphs (**C**,**D**), related to mAb production, show the quadruplicate of titers measured in the most stringent selection step.

**Figure 3 pharmaceutics-14-01985-f003:**
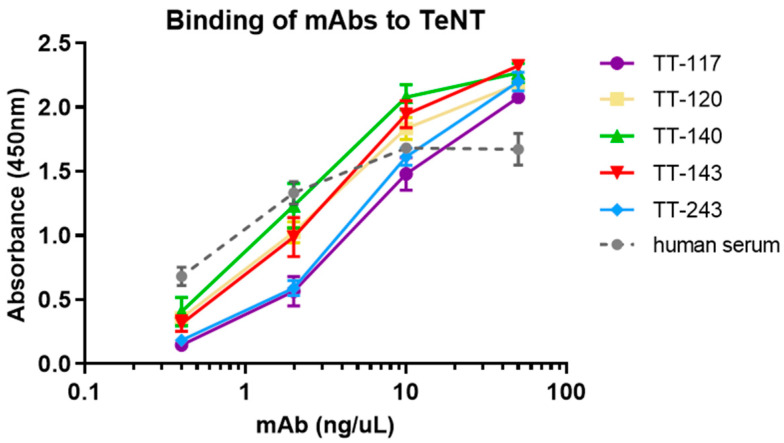
Binding of mAbs to TeNT by ELISA. The mAbs were tested in the 0.4 to 50 ng/mL range. The serum of a vaccinated individual was used as a control in dilutions from 1:2 to 1:16. The graph represents the mean and SD of the absorbance measured in the purified samples of each mAb cultured under four selection conditions.

**Figure 4 pharmaceutics-14-01985-f004:**
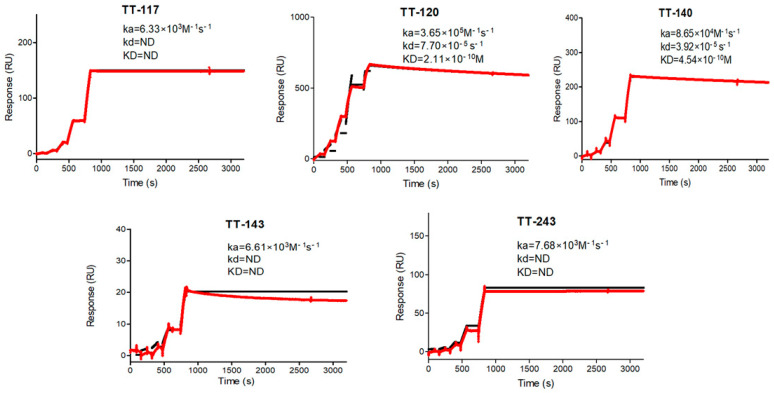
Binding affinity of anti-tetanus mAbs. Affinity constants were determined by single-cycle kinetics using Biacore T200 as described in the Materials and Methods. The red line represents the observed data, and the black line represents the fitted results in a Langmuir 1:1 model. ND = not determined: there was no dissociation in the period of 36,000 s. An increasing concentration of the analyte was applied without dissociation time between them. The graphs show the time up to 3000 s for better visualization.

**Figure 5 pharmaceutics-14-01985-f005:**
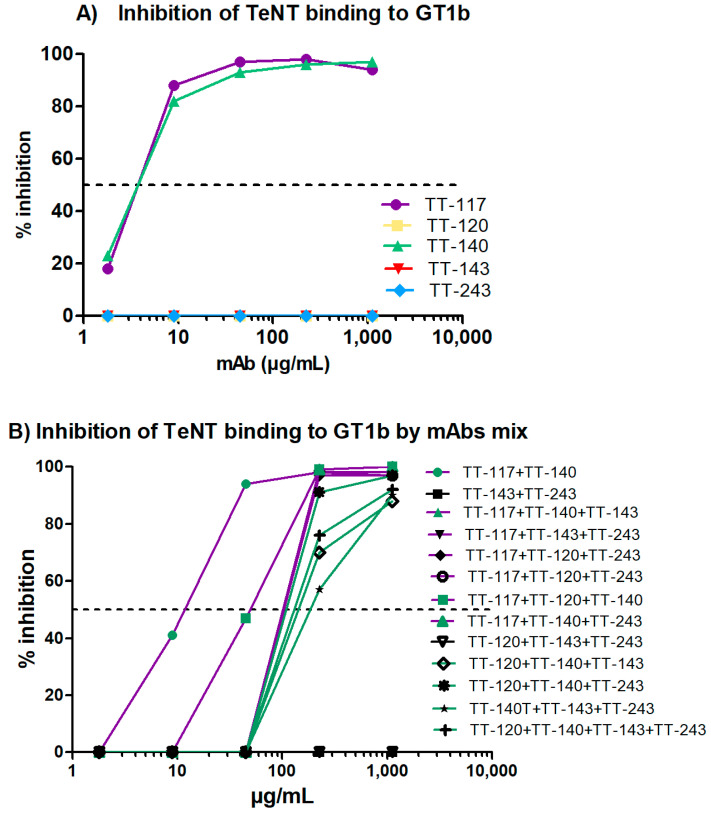
ELISA of the inhibition of binding of TeNT to ganglioside GT1b. The mAbs’ capacity to inhibit TeNT binding was assessed by incubating GT1b with an appropriate level of TeNT alone and increasing the mAb concentration from 0.4 to 50 ng/mL. (**A**) Five mAbs evaluated individually. (**B**) mAbs evaluated in mixtures; the total antibody concentration was always the same. The purple lines represent mixtures containing mAb TT-117; when TT-140 was also present, the symbols are green. The green lines represent mixtures containing only mAb TT-140.

**Figure 6 pharmaceutics-14-01985-f006:**
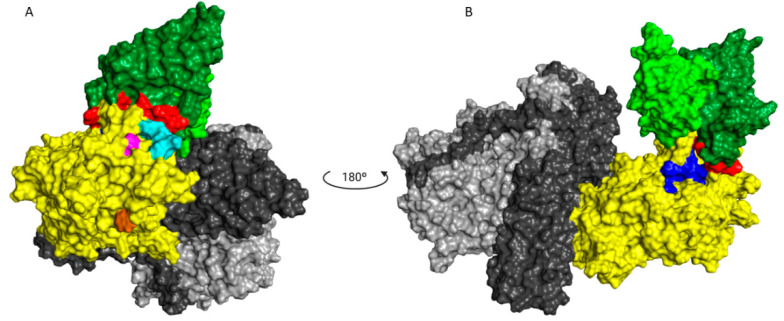
Molecular docking. (**A**) Front of the TT-140-TeNT complex showing the interaction of mAb and epitope predicted by peptide array. (**B**) TT-140-TeNT complex rotated 180° showing the proximity of TT-140 to the predicted epitope of TT-117. In green, the mAb TT-140 (heavy chain in dark green and light chain in light green) with antigen binding residues highlighted in red. In cyan, the TT-140 epitope prediction by the peptide array is shown. The W and R pockets are shown in brown and pink, respectively. Fragment C of TeNT is marked in yellow, the L domain in light grey, and H in dark grey. The TT-117 linear epitope 1119-1128 indicated by the peptide array is shown in blue.

**Figure 7 pharmaceutics-14-01985-f007:**
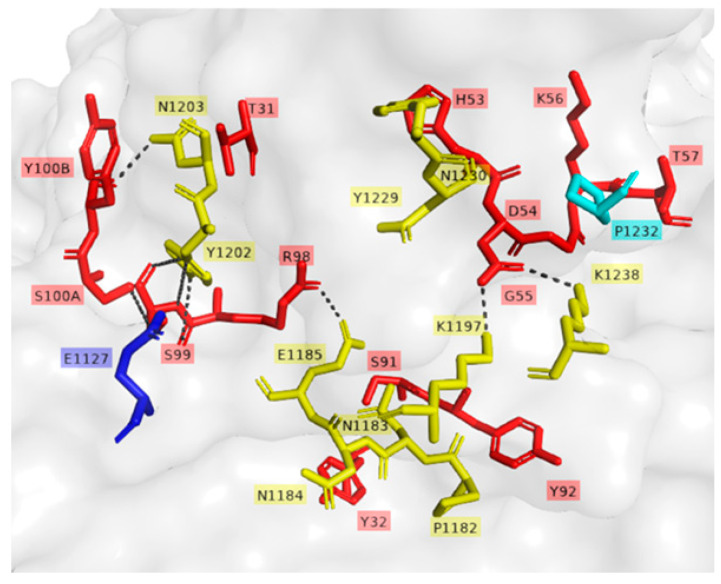
In silico prediction of TT-140 binding to TeNT. Interaction residues of TT-140 and TeNT complex. Representation of hydrophobic and hydrogen bonds’ interaction. Residues of TeNT are marked in yellow and residues of TT-140 are marked in red. TeNT residues of the predicted epitopes by peptide array for TT-140 and TT-117 are marked in cyan and blue, respectively. Black dashed lines represent hydrogen bonds.

**Table 1 pharmaceutics-14-01985-t001:** Ka, kd, and KD(M) were calculated from each mAb by kinetic and steady-state affinity assay.

mAb	Kinetic Affinity			Steady State
	k_a_ (M^−1^.s^−1^)	k_d_ (s^−1^)	K_D_ (M)	K_D_(M)
TT-117	6.33 × 10^3^	ND	ND	2.58 × 10^−7^
TT-120	3.65 × 10^5^	7.70 × 10^−5^	2.11 × 10^−10^	2.41 × 10^−8^
TT-140	8.65 × 10^4^	3.92 × 10^−5^	4.54 × 10^−10^	1.10 × 10^−7^
TT-143	6.61 × 10^3^	ND	ND	2.45 × 10^−7^
TT-243	7.68 × 10^3^	ND	ND	6.08 × 10^−7^

ND = not determined. There was no dissociation in the period of the 36,000 s.

**Table 2 pharmaceutics-14-01985-t002:** Glycan profiles of the anti-tetanus mAbs.

Glycan	TT-117	TT-120	TT-140	TT-143	TT-243
SA	--	4.28	2.91	--	2.76
A2	7.70	9.34	5.69	22.35	8.88
A2F	3.20	7.19	5.92	10.51	5.93
Mn5	3.90	3.51	9.17	3.37	3.17
G0	3.72	4.04	3.80	3.83	4.11
G0F	64.02	52.44	53.58	44.00	56.93
G1/G1′	--	--	--	1.41	0.46
G1F/G1F’	7.04	5.97	4.49	3.16	4.45
G2	2.04	2.35	3.69	--	1.01
G2F	0.54	0.68	3.37	--	--
Fucosylated	74.80	66.27	67.36	57.66	67.30
Galactosylated	9.62	9.00	11.55	4.57	5.91
Sialyted	10.90	20.81	14.52	32.85	17.57
High mannose	3.90	3.51	9.17	3.37	3.17

The glycan proportion was analyzed for each mAb.

## Data Availability

Not applicable.

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
