# Peer review of "Characterization of Neutralizing Human Anti-Tetanus Monoclonal Antibodies Produced by Stable Cell Lines"

_pharmaceutics, 2022, doi:10.3390/pharmaceutics14101985_

Round 1

Reviewer 1 Report

In their manuscript the authors describe the generation and testing of stable CHO lines expressing human anti-TeNT mAbs.

Tetanus is a potential lethal disease despite world-wide vaccination programs being a constant threat to humans. Treatment is based on polyclonal human or animal (mostly equine) antitoxins. However, batch-to-batch variations and high approval costs call for more modern approaches like monoclonal antibodies blocking TeNT action. Thus, their publication of functional blocking mAbs is of interest to the field.

The authors build on their existing cloned and characterised mAbs which were re-cloned in CHO expression vectors. Purified mAbs were tested in TeNT binding (SPR), ganglioside-binding inhibition, Fc receptor binding and glycan-profiling.

Major points:

The reviewer was unable to find the corresponding sequences of the newly generated plasmids including the sequences of the variable regions. Please provide accession numbers for GenBank o.a. Please refer to Deposition of Sequences and of Expression Data in the Author Instructions: “New sequence information must be deposited to the appropriate database prior to submission of the manuscript. Accession numbers provided by the database should be included in the submitted manuscript. Manuscripts will not be published until the accession number is provided.”

Figure 2. As culture conditions affect yield and other parameters. Culture experiments should be independently repeated to show generalized and robust differences. How often where the data repeated and how many technical replicates where analysed per individual data point? Please conduct 3 independent experiments and plot accumulated data including mean and SD.

Figure 3: How often was this repeated? Please include mean and SD.

Table 1: Please provide the raw data (SPR measurements and fit). Either as novel figure or as supplementary figure. Without these it is hard to assess data quality.

Figure 4: same as for Fig. 3. Moreover, should mixes TT-117 and TT-140 with either TT-120 or TT-243 not behave identical as both blocking mAbs (TT-117 + TT-140) are present in the same concentration?

In the discussion the authors state hat only one publication describes a single mAb capable of blocking TeNT. To my knowledge Wang et al. Cell Rep 2021 describes four individual mAbs capable to prevent TeNT action.

It remains unclear to the reviewer why the authors re-cloned their mAbs for stable CHO expression and not used the recombinant material from their HEK expression system. In particular as HEKs are human cells. Transient expression in HEK cells yields usually 10 to 1000 mg mAb which is most often enough for further experiments.

Please compare the SPR measurements for the ‘old’ HEK-derived mAbs with the newly CHO-expressed mAbs. This may be a supplementary figure but confirms that cell line and expression vectors have no impact on antigen recognition.

The authors discuss a possible sterically hindrance of TT-117 and TT-140. Please confirm this by SPR binning experiments or by ELISA competition.

Some groups did already similar experiments and mAbs like TT104, TT0069 or Teth6 look quite promising. Please discuss in more detail how your mAbs compare to published mAbs. E.g. amount of antibody needed for neutralisation, affinity data, epitope regions, and perhaps structural differences.

Minor points:

Line 36: Gram is a name and should be capitalised: “...Gram-positive...”

Line 37: Please use tetanus neurotoxin (TeNT) instead of tetanus toxin (TeNT) as C. tetani can produce other toxins.

Line 61: Why is the damage of TeNT caused irreversible? Patients can recover without long-lasting effects. Also, the cleaved VAMP will eventually fully be replaced.

Line 148: The degree symbol is underlined

Line 154: Please replace 4.7 N by molarity M

Line 192: Could not find the pH of the 0.25 M EDTA solution

Table 1 and throughout manuscript: Replace the x by the multiplicator symbol ×. Replace the dash - by minus –. E.g. 2.58x10-7 -> 2.58×10–7

General comment: If you are going to aim on treating humans would not stable expression in HEK be a better option (human cells more similar glycosylation)?

Author Response

The response to the comments is attached below.

Reviewer 2 Report

Summary:

The investigators previously developed a panel of mAbs that neutralize tetanus toxin. They were obtained by screening the blood of vaccinated humans. A big technological leap was made in the current study when they cloned and expressed the mAbs to get enough for large-scale trials. Thus some of the mAbs were engineered by recombinant technology. Three of the mAbs appear to neutralize the toxin as tested in mice. Two mAbs (117 and 140) appear to prevent binding of the toxin to it eukaryotic target. A number of kinetic, molecular docking and glycan experiments were performed characterizing the mAbs.

General comments:

Overall, this is a well-designed study that is clearly presented. The authors have made significant progress since their study using mice as a model to block TeNT binding. The molecular docking and ganglioside binding experiments were cleverly performed and yield excellent insights. The results clearly show the effect of the TT-117 and TT-140 mAbs (which outperformed the others) and I expect these will be the focus for future clinical trials. 

Because of the high quality of the study, my comments below are more restricted to minor presentation issues rather than scientific concerns.

Major comments:

It seems that although TT-243 is produced at higher levels, it is a slightly weaker binder of the TeNT and does not inhibit toxin binding to the target at all. This is unfortunate, but is a common occurrence when dealing with mAbs. This point could be discussed. The TT-243 was not a codon optimized version, but still was a good producer/secretor, can you explain that?

Minor Comments:

Not sure what the “7,1” and “7,9” represent in lines 56 and 57 of the manuscript. Please clarify.

SPR was never defined in the manuscript.

Figure 1 – cannot read the labels of the color-coded inserts. In general, the text in this figure needs to be increased.

Line 368: Change “elimination” to “eliminated”

Line 378: close the parenthesis

Author Response

The response is attached below.

Round 2

Reviewer 1 Report

Acceptable in the revised version.

Author Response

The manuscript was edited in english, as requested. 
